# Magnetic-Moment-Induced Metal–Insulator Transition in ThMnXN (X = As, P): A First Principles Study

**Smritijit Sen** [1] and **Haranath Ghosh** [2,3,*]

1   Univ. Lille, CNRS, Centrale Lille, ENSCL, Univ. Artois, UMR 8181—UCCS—Unité de Catalyse et Chimie du Solide, F-59000 Lille, France
2   Raja Ramanna Centre for Advanced Technology, Indore 452013, India
3   Homi Bhabha National Institute, Anushakti Nagar, Mumbai 400094, India
*   Correspondence:hng@rrcat.gov.in

**Abstract:** In this work, we show magnetic-moment-induced metal–insulator transitions in ThMnXN (X = As, P) and elucidate some of the experimentally observed results obtained by Zhang et al. through a first principles density functional study. Our calculations revealed that the magnetic ground states of ThMnXN (X = As, P) are C-type anti-ferromagnets with a small energy gap ($\sim$0.4 eV) at the Fermi level, which is in good agreement with the experiments. Our constraint moment calculations revealed local magnetic moments of 3.42 $\mu_B$ and 3.63 $\mu_B$ in Mn atoms for ThMnAsN and ThMnPN, respectively, which are consistent with the experimentally measured local magnetic moment for Mn atoms. An effective Hubbard U = (U − J) of 0.9 eV was applied within the GGA+U formalism in ThMnPN to reproduce the experimental Mn moment. We also found that, as the Mn moments decrease in ThMnXN (X = As, P), the energy gaps also decrease and finally disappear at Mn moment 2.7 $\mu_B$ for ThMnAsN and 2.8 $\mu_B$ for ThMnPN. Therefore, our results stipulate a possible metal–insulator transition in ThMnXN (X = As, P) induced by the Mn local moment.

**Keywords:** magnetism; first principles calculation; electronic structure

## 1. Introduction

ZrCuSiAs-type (1111) Mn-based compounds are of special interest because of their insulating anti-ferromagnetic nature, similar to the parent compounds of high $T_c$ cuprate superconductors. In general, the ZrCuSiAs-type (1111) structure is known for its diverse physical properties such as Pauli paramagnetism (PM) with low-temperature superconductivity in LaNiAsO, itinerant ferromagnetism (FM) in LaCoAsO, and metallic and insulating anti-ferromagnetism (AFM) in LaCrAsO and LaMnPnO, respectively [1–6]. Another addition to the list is ThFeAsN, a high-temperature superconductor at ambient pressure [7]. On the other hand, ThNiAsN is a electron–phonon coupled superconductor with a $T_c$ of 4.3 K [8]. A number of experimental, as well as theoretical studies reveal various interesting physical properties of Th-based 1111 compounds, which include ferromagnetism, strong magnetic fluctuations, superconductivity, Lifshitz topological transitions, etc. [9–12]. ThMnXN (X = As, P) was synthesized recently by Zhang et al. [13]. X-ray and neutron diffraction studies on ThMnXN (X = As, P) revealed the crystal structure and C-type anti-ferromagnetic ordering at 300 K. The magnetic susceptibility curve as a function of temperature showed a cusp at 36 and 52 K, respectively, for ThMnPN and ThMnAsN. These susceptibility cusps are associated with the magnetic transition for $Mn^{2+}$ moments [13]. Like ThFeAsN and ThNiAsN, inbuilt internal chemical pressure, arising from a shorter c-axis (as compared to the other ZrCuSiAs-type 1111 compounds) was also observed in ThMnXN (X = As, P). Resistivity, as well as specific heat data suggest an increase in the density of states near the Fermi level. The experimental trend of resistivity at a low temperature clearly suggests a semiconducting behavior for ThMnXN (X = As, P). In the case of ThMnPN, the resistivity versus temperature plot clearly indicates that, with the increase of temperature, the

metallic behavior increases. The ordered magnetic moments of $Mn^{2+}$ at 4 K are 3.60 $\mu_B$ and 3.41 $\mu_B$ for ThMnPN and ThMnAsN, respectively. However, at 300 K, the ordered magnetic moments of $Mn^{2+}$ are 2.69 $\mu_B$ and 2.30 for ThMnPN and ThMnAsN, respectively [13]. Therefore, the ordered magnetic moments of $Mn^{2+}$ are suppressed at a high temperature. A spin reorientation transition for the $Mn^{2+}$ moment has been reported at a low temperature in NdMnAsO, PrMnSbO, and CeMnAsO, where the direction of the Mn moment changes from the c-axis to the ab plane. In this case, at lower-temperature AFM ordering of the local moment of $Ln^{3+}$ was observed [14–16], whereas no such spin transition or modifications of the Th local moments in ThMnXN (X = As, P) has been reported. It is also noteworthy to mention here that pressure-induced metal–insulator transition is observed in $BaMn_2As_2$ experimentally along with the proclaimed superconductivity [17]. On the other hand, the pressure-induced metal–insulator transition is reported in the ThMnAsN system from first principles simulation [18]. Therefore, it is interesting to see how the electronic structure is affected by the reduction of Mn local moments at a higher temperature.

In this work, we first found the magnetic ground states with accurate local magnetic moments of ThMnXN (X = As, P) consistent with the experiments. In the next step, we performed constrained moment calculation (fixed spin calculation), where we fixed the local moment of $Mn^{2+}$ to a particular value and calculated the density of states for that particular Mn local moment. Our results indicate that, with the lowering of the local Mn moment, both systems become metallic.

## 2. Computational Methods

The crystal structures of ThMnXN (X = As, P) are tetragonal with space group symmetry $P4/nmm$ (Space Group No. 129). ThMnXN (X = As, P) consists of two alternating layers of MnX and ThN. The MnX plane is very similar to the FeAs/Se plane of the Fe-based superconductors, where the Mn atoms are in the same plane, but the As/P atoms are situated above and below of that plane. The height of these As/P atoms from the Mn plane is known as the "anion height". The experimental lattice parameters of tetragonal ThMnXN (X = As, P) at 4 K were used as the input of our first principles density functional theory calculations [13]. Our first principles calculations were performed by employing the projector augmented wave (PAW) method as implemented in the Vienna ab initio simulation package (VASP) [19–21]. The exchange correlation functional was treated under the generalized gradient approximation (GGA) within the Perdew–Burke–Ernzerhof (PBE) functional [22].

We considered four magnetic spin arrangements, namely ferromagnetic (FM), C-type anti-ferromagnetic, A-type anti-ferromagnetic, and G-type anti-ferromagnetic for the magnetic Mn atoms. We constructed a $\sqrt{2} \times \sqrt{2} \times 2$ super-cell in order to implement the above-mentioned spin arrangements. We performed a spin-polarized single-point energy calculation using various spin configurations, as mentioned above. The wave functions were expanded in the plane waves basis with an energy cutoff of 550 eV, and the energy tolerance of the self-consistent calculations was set to $10^{-6}$ eV. The sampling of the Brillouin zone was performed using a Γ-centered $6 \times 6 \times 3$ Monkhorst–Pack grid. We also carried out constrained moment calculations as implemented in VASP, where we constrained both the size and the direction of the magnetic moments. Applying constraints in the magnetic moments added a penalty contribution to the total energy. This contribution decreased with the increase of a parameter called $\lambda$. First, we chose $\lambda = 1$ and, then, increased $\lambda$ stepwise (up to 20) to decrease the penalty contribution to $\sim 10^{-4}$ eV. RWIGS specifies the Wigner–Seitz radius for each atom type. For each atom, we used the values of RWIGS such that the overlap between the spheres was minimized and the sum of the volume of the spheres was close to the total volume of the unit cell. We used the DFT+U approach introduced by Dudarev et al. [23], where the effective Hubbard U was $U_{eff} = U - J$ (U represents the on-site Coulomb repulsion, and J represents the exchange interaction). A number of methods for determining the U parameter from first principles exist in the literature [24–31]. Usually, $U_{eff}$ is determined empirically by varying for values of U

that reproduce the experimental data, such as the band gap, magnetic moment, etc., of a given material. The empirical approach may not be applicable if no experimental data exist, which might be the case, especially in material discovery efforts. Fortunately, for our system, the experimental value of the magnetic moment is available. Therefore, we chose the value of $U_{eff}$ empirically. We gradually increased the value of $U_{eff}$ from 0 to 1 eV and calculated the local magnetic moment of the Mn atom in each case. For $U_{eff} = 0.9$ eV, the local Mn moment was closest to the experimentally measured Mn moment.

## 3. Results and Discussion

First, we found the magnetic ground state of the tetragonal ThMnXN (X = As, P). As mentioned earlier, we considered three different anti-ferromagnetic (AFM) spin arrangements along with the ferromagnetic (FM) spin configuration (see Figure 1). In order to find the magnetic ground state, we performed single-point energy calculations using the experimental lattice parameters, as well as the atomic positions for various spin configurations, namely FM, C-AFM, A-AFM, and G-AFM. The experimental structures and the fully relaxed structures were not very different from each other. For example, in the case of ThMnAsN, changes in the optimized lattice parameters with respect to the experimental lattice parameters were less than 0.2%, whereas we noticed a 1.4% change in the values of experimental $z_{As}$ (z coordinate of As) when the structure was fully relaxed. The Pulay stress was $<10^{-4}$ kB for the experimental structure. The x, y, and z components of the total drift forces were 0, 0, and 0.000255 eV/Å, respectively.

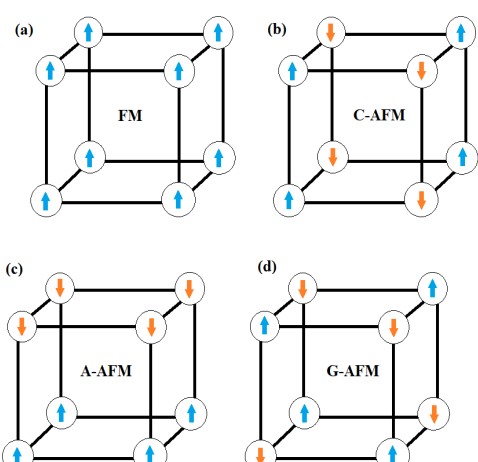

**Figure 1.** Schematic representation of (**a**) ferromagnetic (FM), (**b**) C-type anti-ferromagnetic (AFM), (**c**) A-type anti-ferromagnetic (C-AFM), and (**d**) G-type anti-ferromagnetic (G-AFM) spin arrangements for tetragonal ThMnAsN.

In Table 1, we depict our calculated total energies and local magnetic moments of the Mn atoms of ThMnXN (X = As, P) for the NM, FM, A-AFM, C-AFM, and G-AFM spin configurations. It turned out that, for both systems, the C-AFM spin arrangement had the lowest total energy among all the other spin arrangements that we considered. Therefore, ThMnXN (X = As, P) has a C-AFM ground state, which is consistent with the experimental findings. From Table 1, we clearly see that the magnetic moments in the Mn atoms were 3.52 $\mu_B$ and 3.34 $\mu_B$ for ThMnAsN and ThMnPN, respectively, with the C-AFM spin configuration. The magnetic moments in the Mn atoms, as well as the total energy for the C-AFM and G-AFM spin arrangements in ThMnXN (X = As, P) were quite similar, but differed significantly from those of the A-AFM and FM spin arrangements. Our calculated ground state Mn moments (C-AFM spin configuration) for ThMnAsN was quite close to that of the experimental one (measured at 4 K). However, for ThMnPN, the value of the Mn moments for the C-AFM spin configuration (magnetic ground state) was remarkably lower than the experimentally measured value of the Mn moments at 4 K.

**Table 1.** Calculated total energies (with respect to the NM state) and local magnetic moments of the Mn atoms of ThMnXN (X = As, P) with the experimental structure (at 4 K) for the NM, FM, A-AFM, C-AFM, and G-AFM states. The energy of the NM state was set to zero.

| Magnetic Order | ThMnAsN | | ThMnPN | |
| --- | --- | --- | --- | --- |
| | Energy (meV/f.u.) | Mn Moment ($\mu_B$) | Energy (meV/f.u.) | Mn Moment ($\mu_B$) |
| NM | 0 | 0 | 0 | 0 |
| FM | −317.8 | 2.39 | −332.8 | 2.03 |
| A-AFM | −322.3 | 2.69 | −332.1 | 1.91 |
| C-AFM | −781.1 | 3.52 | −701.7 | 3.34 |
| G-AFM | −780.3 | 3.52 | −700.9 | 3.34 |

Experimental values of the Mn moment of ThMnAsN and ThMnPN were 3.41 $\mu_B$ and 3.60 $\mu_B$, respectively, at 4 K.

We present our calculated density of states (spin-polarized) of ThMnAsN and ThMnPN in Figures 2 and 3, respectively, for all four magnetic spin arrangements.

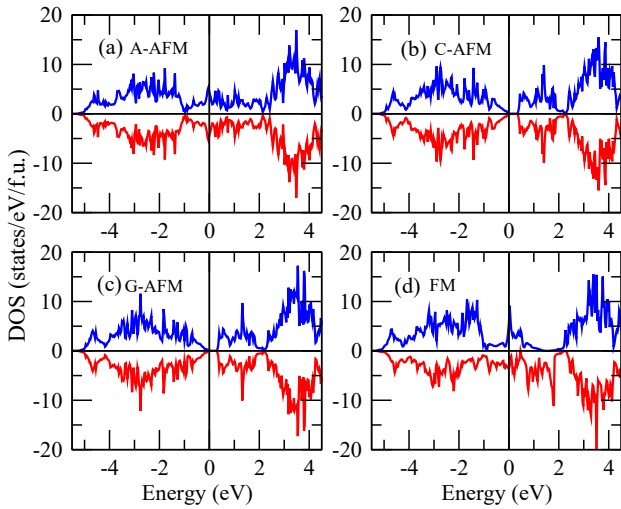

**Figure 2.** Calculated density of states of ThMnAsN for the (**a**) A-AFM, (**b**) C-AFM, (**c**) G-AFM, and (**d**) FM spin arrangements. the up-spin and down-spin density of states are indicated by blue and red lines, respectively. The Fermi level is denoted by a vertical black line at $E = 0$ eV.

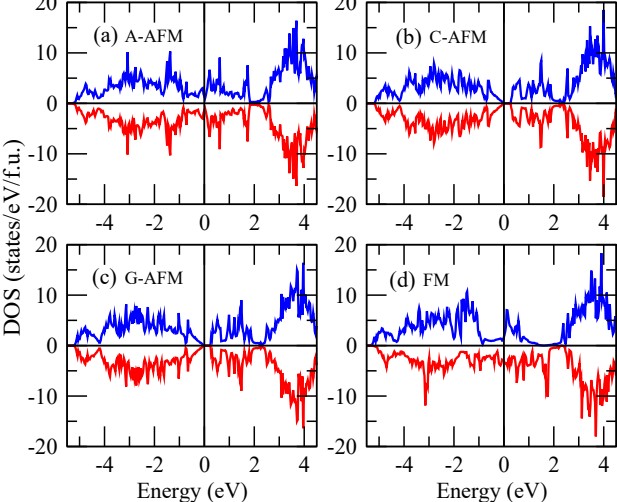

**Figure 3.** Calculated density of states of ThMnPN for the (**a**) A-AFM, (**b**) C-AFM, (**c**) G-AFM, and (**d**) FM spin arrangements. Up-spin and down-spin density of states are indicated by blue and red lines, respectively. The Fermi level is denoted by a vertical black line at $E = 0$ eV.

It is quite evident from Figures 2 and 3 that, in the case of the C-AFM and G-AFM spin configurations, both of these systems possessed a small energy gap in the density of states at the Fermi level (semiconducting behavior at a low temperature). On the other hand, for the other two spin configurations (A-AFM and FM), the density of states clearly suggests a metallic behavior for ThMnXN (X = As, P). We further performed constraint moment calculations for both systems, where we fixed the size, as well as the direction of the local Mn moment along the c-axis. In these calculations, we fixed the magnetic moment of the Mn atoms in ThMnXN (X = As, P) and performed single-point energy calculation for the C-AFM spin arrangement at each fixed Mn local moment. In Figure 4a,b, we depict the variation of the total energy (red square) as a function of local magnetic moment of the Mn atoms for ThMnAsN and ThMnPN, respectively, with the C-AFM spin arrangement.

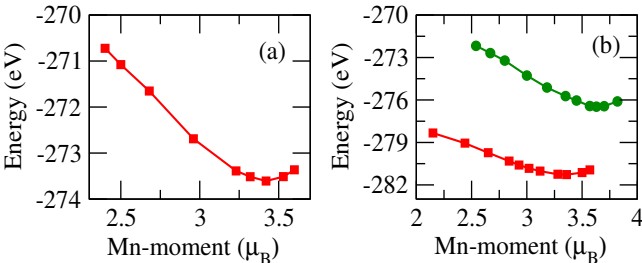

**Figure 4.** Variation of the total energy with the fixed local Mn moment for (**a**) ThMnAsN and (**b**) ThMnPN (green circles indicate the variation of the total energy with the fixed Mn magnetic moment with an effective Hubbard U of 0.9 eV) with the C-AFM spin configuration.

We can see from the total energy versus Mn moment curve of ThMnAsN (Figure 4a) that there exists an energy minimum at around a Mn moment of 3.40 $\mu_B$, which indicates the ground state magnetic moment of the Mn atoms. This value of the Mn moment is in good agreement with the experimentally observed value of the Mn moment at a low temperature (4 K) in ThMnAsN. However, we see from Figure 4b that the value of the Mn moment ($\sim$3.30 $\mu_B$) at the total energy minimum for ThMnPN is significantly smaller than that of the experimental value of the Mn moment measured at 4 K (3.60 $\mu_B$) [13]. Therefore, for ThMnPN, we performed the GGA+U calculation [32] and tuned the value of the Hubbard U such that it produced the experimental value of the Mn moment, i.e., 3.60 $\mu_B$. We found that, with the value of effective Hubbard U = 0.9 eV in the Mn d orbitals for ThMnPN, we can reproduce the experimentally measured Mn moment at 4 K. In Figure 4b, we display the variation of the total energy (green circle) with the fixed Mn moment in ThMnPN in the presence of an effective Hubbard U = 0.9 eV. We found that, with the introduction of U, the energy minimum shifted to a value 3.63 $\mu_B$, which is very close to the experimentally measured value (3.60 $\mu_B$) of the Mn moment in ThMnPN at 4 K. It also is noteworthy to mention here that the experimental study indicated a magnetic phase transition for the $Mn^{2+}$ moments in both materials. The experimentally measured local Mn moments at a temperature of 4 K and 300 K for ThMnAsN were 3.41 $\mu_B$ and 2.30 $\mu_B$, respectively. On the other hand, the local Mn moments in ThMnPN at a temperature of 4 K and 300 K were 3.60 $\mu_B$ and 2.69 $\mu_B$, respectively. Therefore, we calculated the density of states of ThMnXN (X = As, P) for the C-AFM spin configuration at various fixed Mn moments to see the modifications of low-energy electronic structures with Mn moments. In Figures 5 and 6, we display the density of states of ThMnAsN and ThMnPN for various fixed Mn moments.

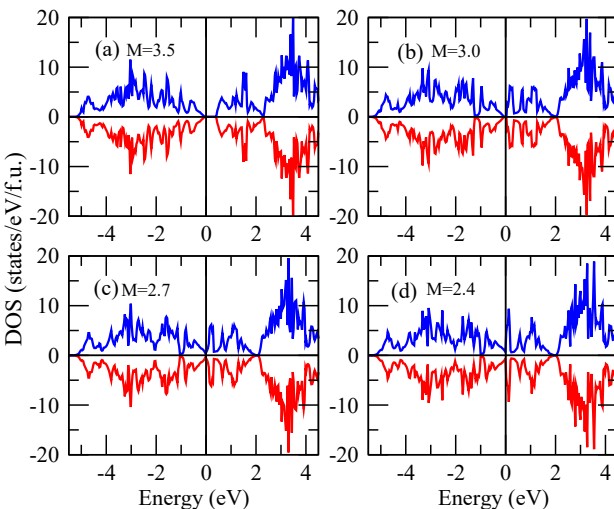

**Figure 5.** Calculated density of states of ThMnAsN with the C-AFM spin configuration for fixed Mn moments of (**a**) M = 3.5 $\mu_B$, (**b**) M = 3.0 $\mu_B$, (**c**) M = 2.7 $\mu_B$, and (**d**) M = 2.4 $\mu_B$. Up-spin and down-spin density of states are indicated by blue and red lines, respectively. The Fermi level is denoted by a vertical black line at $E = 0$ eV.

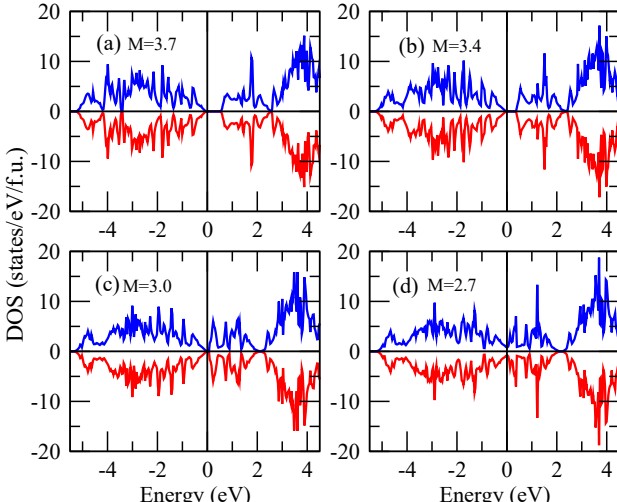

**Figure 6.** Calculated density of states of ThMnPN with the C-AFM spin configuration for fixed Mn moments with an effective Hubbard U of 0.9 eV (**a**) M = 3.7 $\mu_B$, (**b**) M = 3.4 $\mu_B$, (**c**) M = 3.0 $\mu_B$, and (**d**) M = 2.7 $\mu_B$. Up-spin and down-spin density of states are indicated by blue and red lines, respectively. The Fermi level is denoted by a vertical black line at $E = 0$ eV.

In the case of ThMnPN, the density of states was calculated using an effective Hubbard U of 0.9 eV within the GGA+U formalism. It is quite apparent from Figures 5 and 6 that, at a higher Mn moment, the density of states at the Fermi level possessed an energy gap, but at the lower values of the Mn moment, this energy gap vanished and became metallic. From Figures 5d and 6d, we found that a significant amount of electronic states was present at the Fermi level for both systems at the lower values of fixed local Mn moments. In Figure 7a,b, we depict the variation of the energy gap as a function of the Mn moment for ThMnAsN and ThMnPN, respectively. For ThMnAsN and ThMnPN, the energy gaps disappeared at a Mn moment around 3 $\mu_B$ and 2.9 $\mu_B$ (2.80 $\mu_B$ with U = 0.9 eV). Thus, our results indicate that a metal–insulator transition occurs due to the collapse of the local Mn magnetic moment in ThMnXN (X = As, P), which explains the origin of the magnetic susceptibility cusps observed at 36 and 52 K, respectively, for ThMnPN and ThMnAsN in the experiments [13]. Furthermore, it also explains the abnormal increase of the density of states at the Fermi

level as is evident from the resistivity and specific heat measurements [13]. We found an energy gap of around 0.3 eV (for a fixed Mn moment of 3.4 $\mu_B$) and 0.4 eV (for a fixed Mn moment of 3.6 $\mu_B$) for the C-AFM ground state of ThMnAsN and ThMnPN, respectively. Clearly, the energy gap of ThMnPN is higher than the energy gap of ThMnAsN, and the energy gap of ThMnPN vanishes at a higher Mn moment compared to that of ThMnAsN. This certainly validates the magnetic susceptibility cusps of ThMnPN at a temperature (36 K) lower than that of ThMnAsN (magnetic susceptibility cusps at 52 K) [13]. However, further experimental studies measuring the temperature-dependent structural parameters, as well as local magnetic moments are required to elucidate the metal–insulator transitions in ThMnXN (X = As, P). The anion height would vary with temperature, which may be a reason for the decrease in the magnetic moment. Therefore, further experimental and theoretical studies are also required to understand the role of the electronic correlation in the magnetism of these two systems.

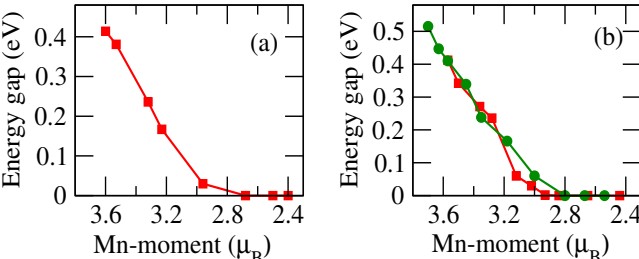

**Figure 7.** Variation of the energy gap at the Fermi level at different fixed Mn magnetic moments for (**a**) ThMnAsN and (**b**) ThMnPN (green circles indicate the energy gap at various fixed Mn magnetic moments with an effective Hubbard U of 0.9 eV) with the C-AFM spin configuration.

## 4. Conclusions

In this section, we briefly summarize our results. Through our first principles calculations, we found the magnetic ground state of ThMnXN (X = As, P), and it turned out to be anti-ferromagnetic with the C-type spin orientation. Our results are in good agreement with the experimental findings [13]. A small energy gap (∼0.4 eV) was observed in the density of states near the Fermi level for both systems. Our constraint moment calculations revealed local magnetic moments of 3.42 $\mu_B$ and 3.63 $\mu_B$ in Mn atoms, respectively, for ThMnAsN and ThMnPN. However, in the case of ThMnPN, we implemented the GGA+U approach to reproduce the experimental value of the local Mn moment. We also found that, as the Mn moments decrease in ThMnXN (X = As, P), the energy gaps also decrease and eventually disappear at a certain value of the local Mn moment (2.7 $\mu_B$ for ThMnAsN and 2.8 $\mu_B$ for ThMnPN). Therefore, our results stipulate a magnetic-moment-induced metal–insulator transition in ThMnXN (X = As, P) for the very first time.

**Author Contributions:** S.S.: data curation, writing—original draft preparation, methodology, software. H.G.: conceptualization, writing—reviewing and editing, supervision. All authors have read and agreed to the published version of the manuscript.

**Funding:** This research received no external funding.

**Institutional Review Board Statement:** Not applicable.

**Informed Consent Statement:** Not applicable.

**Data Availability Statement:** Data will be available upon request.

**Conflicts of Interest:** The authors declare no conflict of interest.

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
