# Peer review of "Magnetic-Moment-Induced Metal–Insulator Transition in ThMnXN (X = As, P): A First Principles Study"

_magnetochemistry, doi:10.3390/magnetochemistry9010016_

Round 1
Reviewer 1 Report
In the manuscript under review the authors show magnetic moment induced metal-insulator transitions in ThMnXN (X=As, P) through first principles density functional study. Energies and magnetic moments for the experimental structures (at 4 K) with NM, FM, A-AFM, C-AFM, and G-AFM states were calculated. For both of the ThMnXN (X=As, P) structures C-AFM spin arrangement has the lowest total energy among all other spin arrangements. The value of Mn spin obtained for ThMnAsN with first-principle calculations is in a good agreement with the experiment whereas Mn spin for ThMnPN is underestimated in comparison with the experiment. An effective Hubbard U of 0.9 eV is applied within the GGA+U formalism in ThMnPN to reproduce the experimental Mn moment. The authors also calculated the variation of energy gap at the Fermi level at different fixed Mn magnetic moments for ThMnAsN and ThMnPN with C-AFM spin configuration. It was found that energy gaps disappear at Mn spin around 3 muB and 2.8 muB, respectively. The results indicate that a metal-insulator transition occurs due to the collapse of local Mn magnetic moment which explains the origin of magnetic susceptibility cusps observed at 36 and 52 K respectively for ThMnPN and ThMnAsN in the experiments.
The manuscript is very clear and easy readable. The results are clearly presented. Several points should be considered by the authors before the manuscript can be accepted for publication in Magnetochemistry.
1. It would have been better to clarify in abstract that the authors not only carry out first principle calculations, but also interpret some experimental results. For the moment it is only written that “an effective Hubbard U of 0.9 eV is applied within the GGA+U formalism in ThMnPN to reproduce the experimental Mn moment”.
2. How was an effective Hubbard U calculated? Did the authors use random phase approximation? I am suggesting adding the details of this calculation.
3. How did the authors choose the parameters for VASP spin-constraint calculations (e.g., the weight LAMBDA before the penalty term, RWIGS, etc.)? It would have been better to add these details in the manuscript.
4. The authors are using the experimental structures obtained at 4 K for calculations. How are these structures far from the fully relaxed structures (w.r.t. positions and unit cell)? What are the values of forces and stresses calculated for these experimental structures?
Reviewer 2 Report
This work based on first principle calculations shows possible ground states and metal-insulator transitions in ThMnXN (X=As, P) system, which is a new candidate for possible superconductivity, similar to the 1111-type iron-based superconductors. Indeed, there are only a few researches on this family, as the authors cited in Ref.8-13. The magnetic ground state was reported as C-type anti-ferromagnetic structure, same as the parent compound of 1111-type iron-based superconductors. This work give the details on the magnetism, where there is a small energy gap (∼ 0.4 eV) at the Fermi level, a local magnetic moment around 3.5 muB, and a possible metal-insulator transition induced by Mn local moment. Such results will certainly help the community to pay attention on this compound, which may show close behaviors to cuprates. I find this manuscript is well presented and well written. Therefore, I would suggest to publish as it is.
Author Response
The reviewer has accepted the manuscript as it is.